# One for All: Multi-Domain Joint Training for Point Cloud Based 3D Object Detection

**Zhenyu Wang**[1]    **Yali Li**[1] *    **Hengshuang Zhao**[2] *    **Shengjin Wang**[1]

[1] Department of Electronic Engineering, Tsinghua University, BNRist
[2] The University of Hong Kong
{wangzy20@mails., liyali13@, wgsgj@}tsinghua.edu.cn, hszhao@cs.hku.hk

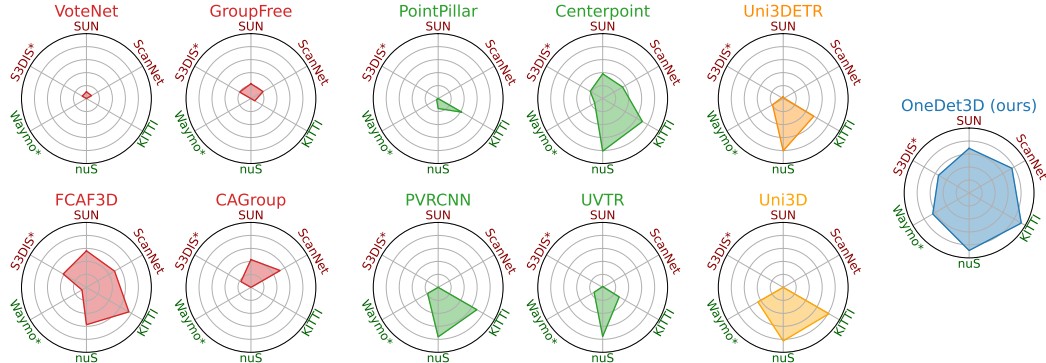

Figure 1: The high-level overview comparing the multi-domain joint training performance of 10 existing 3D detectors and our OneDet3D. These models are jointly training on the indoor datasets SUN RGB-D (SUN), ScanNet, and outdoor datasets KITTI, nuScenes (nuS). We also evaluate the cross-domain performance on the indoor S3DIS and outdoor Waymo datasets. The center of the circle means that the corresponding metric is less than 10%, and the outermost means 90%. Existing indoor detectors are plotted in red, outdoor detectors are in green, and detectors that aim for different scenes are in orange. Our model has the remarkable capacity to generalize across a wide range of diverse 3D scenes (a larger polygon area) with only **one set of parameters** and the same architecture.

## Abstract

The current trend in computer vision is to utilize one universal model to address all various tasks. Achieving such a universal model inevitably requires incorporating multi-domain data for joint training to learn across multiple problem scenarios. In point cloud based 3D object detection, however, such multi-domain joint training is highly challenging, because large domain gaps among point clouds from different datasets lead to the severe domain-interference problem. In this paper, we propose **OneDet3D**, a universal one-for-all model that addresses 3D detection across different domains, including diverse indoor and outdoor scenes, within the *same* framework and only *one* set of parameters. We propose the domain-aware partitioning in scatter and context, guided by a routing mechanism, to address the data interference issue, and further incorporate the text modality for a language-guided classification to unify the multi-dataset label spaces and mitigate the category interference issue. The fully sparse structure and anchor-free head further accommodate point clouds with significant scale disparities. Extensive experiments demonstrate the strong universal ability of OneDet3D to utilize only one trained model for addressing almost all 3D object detection tasks (Fig. 1).

---

*Corresponding authors.

38th Conference on Neural Information Processing Systems (NeurIPS 2024).

# 1 Introduction

3D point cloud based object detection aims to predict the oriented 3D bounding boxes and the corresponding semantic category tags for the real scenes given a point set. Unlike mature 2D detectors [29, 14, 38, 4], which once trained, can generally conduct inference on different types of images in various scenes and environments, current 3D detectors still follow a single-dataset training-and-testing paradigm, *i.e.*, point clouds used during inference should be from the totally same domain as that used during training. Whether indoor [25, 51, 44, 39] or outdoor [45, 16, 35, 32], existing point cloud based 3D detectors can only be trained on datasets from one specific domain, then be tested on the same domain data. Such restriction of training and testing on a single dataset severely hampers the generalization ability of 3D detectors, resulting in a significant lag in the progress of 3D detection compared to 2D in terms of universality.

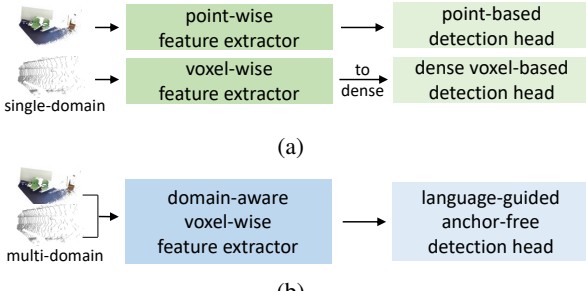

Figure 2: **Illustration of existing 3D detectors (a) and ours (b).** Existing detectors can be divided into point-based (up) and voxel-based (down). Our model has the capacity for joint training on multi-domain point cloud data.

To address this issue, multi-domain joint training (*i.e.*, multi-dataset joint training) should be introduced into point cloud based 3D object detection, to allow 3D detectors to learn from point clouds of different domains through large-scale joint training. In this way, a 3D detector, once trained, can well generalize across various domains of point clouds. The ultimate goal is to obtain a 3D detector that can support unified 3D object detection across different domains with only **one set of parameters**, thereby achieving the target of universal 3D object detection.

The motivation of multi-domain joint training is to learn universal 3D knowledge by leveraging point clouds from different sources and domains, thereby establishing a general representation from 3D data to 3D spatial positions. Through this, a model independent of point cloud source, collection, and domain can be achieved. With common 3D knowledge from diverse point clouds, it can effectively serve as a universal 3D detector and function as a 3D foundation model. However, achieving this is highly challenging and difficult. As can be seen in Fig. 1, due to the significant domain gaps (*e.g.*, point cloud ranges, scenes, object sizes, sparsity, *etc.*), existing 3D detectors fail to support this. Specifically, current 3D detectors can be generally divided into point-based and voxel-based ones. For point-based 3D detectors (the upper part of Fig. 2a) [25, 51, 21], it is difficult to apply the same sampling and grouping technique for different domain data. For voxel-based 3D detectors (the lower part of Fig. 2a) [45, 47, 17, 41], which usually require converting from sparse to dense features for 3D box prediction, the scale differences between indoor and outdoor point clouds make it difficult to represent them using dense features of the same size. This consequently limits existing models to learning domain-specific knowledge, restricting their ability to acquire generalized 3D knowledge.

In this paper, we propose **OneDet3D**, a unified point cloud based 3D detector with only one set of parameters through multi-domain joint training. As in Fig. 2b, we employ 3D sparse convolution for feature extraction, which is more robust to domain gaps compared to point-based feature extractors [26, 27], making it well-suited for adapting to point clouds from different domains. Subsequently, we utilize an anchor-free detection head, where objects are represented by center points [38, 47, 30], enabling direct compatibility with sparse convolution and avoiding the constraints of fixed-size dense features. Such a fully sparse structure, together with the anchor-free detection head using center point representation, provides an effective architecture for multi-domain joint training.

Based on the model architecture, during multi-domain joint training, the domain-interference issue should be further addressed. This issue primarily comprises two aspects: data-level interference caused by differences in point clouds themselves, and category-level interference caused by label conflict among categories across different domains. To mitigate the *data-level interference*, we employ domain-aware partitioning, which partitions parameters where the interference problem mainly exists to be domain-specific and keeps the vast majority shared among different domains. The data-level interference can thus be effectively prevented without increasing the model complexity

too much. Specifically, we partition re-scaling in normalization layers to maintain the consistency of the data scatter, and parameters about context learning for reducing the effect of range disparities. They are guided by a domain router implemented by a domain classifier. To alleviate *category-level interference*, we employ language-guided classification, leveraging the text modality to alleviate conflict issues. We utilize a combination of fully connected layers and sparse convolution for class-specific and class-agnostic classification to ensure compatibility with the anchor-free head.

Our main contributions can be summarized as follows:

- We propose OneDet3D, a multi-domain point cloud joint training model for universal 3D object detection. To the best of our knowledge, this is the first 3D detector that supports point clouds from domains in both indoor and outdoor simultaneously with only *one set of parameters*.
- We propose the domain-aware partitioning in scatter and global context, guided by the domain routing mechanism. In this way, the data-level interference issue caused by point cloud disparities can be alleviated during multi-dataset joint training.
- We integrate the text modality into the anchor-free head classification. Through employing both fully connected layers and 3D sparse convolution for the dual-level of class-agnostic and class-specific classification, the issue of category-level interference can be mitigated.

Extensive experiments demonstrate the one-for-all ability of our OneDet3D. OneDet3D possesses the strong generalization ability in both category and scene, thus effectively achieving the goal of universal 3D object detection. In the close-vocabulary setting, it achieves comparable performance using only one set of parameters. In the open-vocabulary setting, it obtains more than 7% performance.

## 2  Related Work

**3D object detection** aims to predict category tags and oriented 3D bounding boxes for the scene. We primarily discuss methods where point clouds serve as the input. Current 3D detectors can be generally categorized into point-based and voxel-based methods. Point-based methods [25, 51, 21, 34] usually extract point-wise features, then perform clustering and classification for detection. Voxel-based methods [30, 39, 32, 9, 31, 47, 33] usually extract voxel-wise features using 3D sparse convolution, then convert them into dense 3D features for 3D box prediction. Considering the disparities in point clouds, existing 3D detection methods are also separated into indoor 3D detectors [25, 51, 30, 39] and outdoor ones [45, 35, 32, 9, 31, 5, 33], where totally different model architectures are utilized for each. Recently, [41] proposes a unified model architecture for both indoor and outdoor 3D detection. However, these methods still follow a single-dataset training-and-testing paradigm, and cannot address 3D detection for point clouds from various domains with one set of parameters.

**Multi-dataset training** aims to involve multiple datasets from various domains in training, so that the model can generalize in multi-domain data at the inference time. Since RGB images mainly differ in content, while the structural differences in images themselves are not significant, multi-dataset training has been widely studied in the field of 2D object detection [40, 52, 55, 24, 42]. In comparison, substantial differences inherently exist in the point clouds themselves, making multi-dataset training more challenging in the 3D object detection task. Some recent works [48, 43, 46] have studied this problem. However, they only address multi-dataset training within either indoor or outdoor scenes and cannot handle multiple datasets simultaneously from both indoor and outdoor scenes. For example, [40] and [12] deal with multi-dataset training with RGB images, [20] and [48] focus on outdoor-only multi-dataset training, where the discrepancies between different datasets are far less than those between indoor and outdoor point clouds. OneDet3D demonstrates that despite these substantial differences, 3D detection can still be addressed with a universal solution. This is a crucial advancement for generalization in the 3D domain.

## 3  Preliminary

Given a point cloud $x$, 3D object detection aims to predict its label $y$, which consists of the category tags and 3D bounding boxes. Multi-domain (*i.e.*, multi-dataset) data are utilized during training. Denote the domain as $\mathcal{D}$ and the total number of domains as $N$, the total training data can thus be denoted as $\mathcal{D} = \{\mathcal{D}_n = \{(x^{(n)}, y^{(n)})\}\}_{n=1}^{N}$. The purpose of multi-domain joint training is to train a unified model from all these domains, which can obtain the minimum prediction error on all different domains $\mathcal{D}$. The obtained 3D detector should also generalize well on new domains.

Table 1: **Overview of 3D object detection dataset difference**. L, W, and H represent the length, width, and height. For ScanNet and S3DIS, we perform global alignment then measure the range.

| datasets | sensor | point range | | | scene | view |
|---|---|---|---|---|---|---|
| SUN RGB-D [36] | RGB-D camera | L=[-3.2, 3.2]m, | W=[-0.2, 6.2]m, | H=[-2.0, 0.56]m | indoor | front |
| ScanNet [8] | reconstructed | L=[-6.4, 6.4]m, | W=[-6.4, 6.4]m, | H=[-0.1, 2.46]m | indoor | 360° |
| S3DIS [1] | reconstructed | L=[-9.6, 9.6]m, | W=[-9.6, 9.6]m, | H=[0.0, 4.8]m | indoor | 360° |
| KITTI [11] | 64-beam LiDAR | L=[0.0, 70.4]m, | W=[-40.0, 40.0]m, | H=[-3.0, 1.0]m | outdoor | front |
| nuScenes [2] | 32-beam LiDAR | L=[-51.2, 51.2]m, | W=[-51.2, 51.2]m, | H=[-5.0, 3.0]m | outdoor | 360° |
| Waymo [37] | 64-beam LiDAR | L=[-75.2, 75.2]m, | W=[-75.2, 75.2]m, | H=[-2.0, 4.0]m | outdoor | 360° |

In 3D object detection, the following two-level interference exists among different domain point clouds, making multi-domain joint training highly challenging:

**Data-level interference.** As in Tab. 1, it can be observed that sensors for collecting indoor and outdoor point clouds exhibit fundamental differences, resulting in significant disparities in the range covered by the point clouds, with differences exceeding 10 to nearly 20 times. This also leads to substantial differences in object sizes and sparsity within the scenes. Because of such scale differences, it is challenging to utilize the same point-wise clustering technique or feature map with the fixed size during joint training for point clouds from different scenes. Even among datasets that belong to the same category of indoor or outdoor point clouds, there are still slight differences in the sensors used for collection. For instance, SUN RGB-D [36] points are from RGB-D camera captures, while ScanNet points are reconstructed from RGB images. Distinction in the number of LiDAR beams also leads to differences in point cloud sparsity. We thus propose domain-aware partitioning in section 4.2, guided by the routing technique, to alleviate such data-level interference.

**Category-level interference.** Different datasets typically possess distinct label spaces. An object classified as background in one dataset might be considered as foreground in another. Even for the same category, different datasets sometimes employ different classification and definition ways, such as the definition of the "car" category in outdoor datasets. Such dataset-specific taxonomy and annotation inconsistencies pose challenges in unifying multiple label spaces. The category-level differences thus result in the interference problem among multiple datasets during training. We propose the language-guided classification in section 4.3 to mitigate such category-level interference.

## 4 Method

The overview of our OneDet3D is illustrated in Fig. 3. We utilize 3D sparse convolution for feature extraction and anchor-free detection head for 3D box prediction. Based on it, we propose the domain-aware partitioning during feature extraction to alleviate data-level interference, and propose the language-guided classification in the anchor-free head to mitigate category-level interference.

### 4.1 Multi-Domain Joint Training

**Architecture.** We design the architecture of OneDet3D from the feature extractor and the detection head aspects. For the feature extractor, we utilize 3D sparse convolution to extract voxel-wise features. Compared to point-wise structures, voxel-wise features are more robust to domain gaps and less sensitive to hyper-parameters, suitable for multi-domain training. Additionally, sparse convolution is not only computationally efficient but also operates solely on points, thus not relying on fixed-size feature maps. This enables to extract domain-invariant 3D features for multi-domain joint training.

For the detection head, we adopt the anchor-free way, where objects are represented by their center points. It directly regards points from sparse convolutions as centers to represent objects, avoiding the need for conversion from sparse to dense feature maps. We do not employ any pruning layers [13, 30]. Instead, we retain all points until the final stage for box prediction. This helps avoid the issue of requiring different pruning strategies due to variations in point clouds. Such a fully sparse architecture well accommodates point clouds from multiple domains thus serves for multi-domain training.

**Joint training.** During training, due to the disparities in object sizes across different point clouds, the localization accuracy requirements vary greatly among datasets. Considering this, besides classification, regression, and centerness prediction learning, we also introduce the 3D IoU prediction learning to ensure that the box scores accurately represent their positional accuracy.

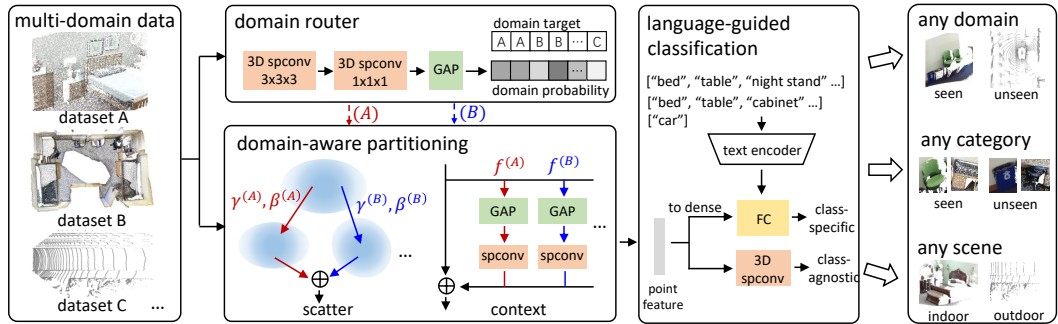

Figure 3: **The overview of OneDet3D.** It utilizes multi-domain point clouds for training. The domain-aware partitioning in scatter and context avoids the data-level interference issue, and the language-guided classification addresses the issue from category-level interference. Once trained, OneDet3D has the one-for-all ability to generalize to unseen domains, categories, and diverse scenes.

For classification, we use soft focal loss as in [41]. For easier optimization, we utilize the IoU in the BEV space, $\text{IoU}_{BEV}$, as the soft target. Specifically, denote the binary target class label as $c$, the predicted class probability as $\hat{p}$, the classification loss is:

$$L_{cls} = -\hat{\alpha}_t \cdot |c \cdot \text{IoU}_{BEV} - \hat{p}|^\xi \cdot \log(|1 - c - \hat{p}|) \tag{1}$$

where $\hat{\alpha}_t = \alpha \cdot c \cdot \text{IoU}_{BEV} + (1-\alpha) \cdot (1 - c \cdot \text{IoU}_{BEV})$. This classification loss can be viewed as using the soft target $\text{IoU}_{BEV}$ in focal loss [19]. By employing $\text{IoU}_{BEV}$, the network classification focuses solely on the position on the horizontal plane, which helps improve the calibration of classification scores. Here, we discard positional information in the height direction to prevent optimization from becoming overly complex, which makes the network easier to converge during joint training.

For regression, we use the 3D IoU loss [53], optimizing with the usual 3D IoU. For both centerness and IoU prediction, we utilize the binary cross entropy loss. The IoU prediction branch is also supervised with the usual 3D IoU. Since different datasets vary in scale, we employ dataset-aware sampling during joint training: sampling datasets first and then randomly selecting samples. Denote network parameters as $\theta$, the objective of network training can thus be formulated as:

$$\arg\min_\theta \sum_{n=1}^{N} \frac{1}{|\mathcal{D}_n|} \sum_{(x^{(n)}, y^{(n)}) \in \mathcal{D}_n} L_{cls}^{(n)} + L_{reg}^{(n)} + L_{centerness}^{(n)} + L_{iou}^{(n)} \tag{2}$$

### 4.2 Domain-Aware Partitioning

During multi-domain joint training, we first aim to mitigate the data-level interference caused by differences in the inherent structure of point clouds. We identify two primary sources of interference. First, due to significant differences between data, interference mainly arises in the normalization layers, which adjust the scatter of data to maintain their consistency. Second, convolution mainly focuses on local information, leading to interference in context learning across different domain point clouds, where scale difference mainly exists. Therefore, we partition parameters about these two aspects into domain-specific ones. We design a domain router to guide such domain-aware training. In this way, the partitioned parameters are responsible for learning domain-equivalent knowledge, while the majority of the model can avoid interference and learn domain-invariant 3D knowledge. This allows multi-domain joint training to effectively acquire universal 3D representations.

**Domain router.** Given the input point cloud $x^{(n)}$, the domain router aims to guide its path for the domain-aware partitioning. We utilize a domain classifier for the routing mechanism by classifying its correct domain label $n$. To achieve this, we employ 3D sparse convolutions with kernel sizes of 3 and 1 for simple feature extraction, then utilize global average pooling (GAP) to obtain the feature of the whole scene. After applying softmax, we obtain the domain probability $\{p_d^{(n)}\}_{n=1}^N$ and directly use cross entropy loss for classification. Due to the large domain differences, this classification task is relatively simple thus the domain router can converge rapidly. During inference, when encountering unseen domain data, such domain probability can indicate its similarity to seen domains and provide its data flowing path, enabling the model to generalize to unseen domains.

**Scatter partitioning.** The normalization layers conduct regularization for input data, thus reducing hierarchical differences in data and making the network easier to train. Then normalization layers conduct re-scaling to adjust the scatter of data. Considering the significant differences in different domains, the same re-scaling operations will lead to disparities in the output data scatters. In this situation, we partition scaling and shifting parameters after normalization for each domain data, so that data scatter in different domains can be partitioned. All other convolution layers can be shared, only scaling and shifting parameters being domain-specific and differing. For unseen domains at the inference time, we introduce the domain probability from the domain router. Specifically, we keep $N$ sets of scaling and shifting parameters $\{\gamma^{(n)}\}_{n=1}^{N}$, $\{\beta^{(n)}\}_{n=1}^{N}$. Scatter partitioning can thus be formulated as:

$$\overline{x} = \sum_{n=1}^{N} p_d^{(n)} \cdot \left[ \frac{x - \mathrm{E}(x)}{\sqrt{\mathrm{Var}(x) + \epsilon}} \cdot \gamma^{(n)} + \beta^{(n)} \right] \tag{3}$$

where we utilize $\overline{x}$ to denote the output of normalization layers. Through this, the network can apply individualized re-scaling operations from different domains, resulting in domain-specific scatter thus effectively mitigating the data-level interference. Only introducing $N$ sets of scaling and shifting parameters almost negligibly increases the model size.

**Context partitioning.** In addition, we separately learn global context information for different domain data to prevent interference between them in terms of global context. Specifically, for features $f$ from the blocks in the feature extractor, we first apply a global average pooling layer to extract the feature of the whole scene, then utilize a 3D sparse convolution to learn its context information. According to previous work [41], global information mainly matters in indoor scenes. We thus only impose context learning for indoor domains. The process of context partitioning can thus be formulated as:

$$\hat{f} = f + \sum_{i \in \mathrm{indoor}} p_d^{(i)} \cdot \mathrm{conv}^{(i)}(\mathrm{GAP}(f)) \tag{4}$$

where we utilize $\hat{f}$ to denote the updated features with partitioned domain-aware global context.

## 4.3 Language-Guided Classification

We then aim to alleviate category-level interference among domains caused by label conflicts. Different datasets inherently possess different label spaces, which leads to the problem of annotation inconsistency. Moreover, at the inference time, unseen domains may involve a label space that is different from those seen during training. Such a category-level difference results in different definitions of the same object, leading to the conflict and interference problems during training. To address this, we utilize language vocabulary embeddings from CLIP [28] for classification. Specifically, we use the prompt "a photo of {name}" to extract language embeddings of the category names from different datasets using CLIP. These language embeddings are then used as parameters of the fully connected layer to perform the final classification, and are kept frozen during training. Each dataset utilizes its own language embeddings, effectively mitigating such interference.

Due to the fully convolution architecture and the anchor-free head, the final classification is typically achieved through 3D sparse convolutions. To incorporate language embeddings, we convert the sparse features of points into dense features, and then use language embeddings for classification through fully connected layers. However, this conversion from sparse to dense features, together with the frozen language embeddings, poses obstacles to gradient backpropagation, making it difficult for the network to converge. To address this, we introduce a class-agnostic classification branch that performs only foreground-background binary classification. This branch is shared across different datasets and implemented using 3D sparse convolution. In this way, a part of classification can be addressed through 3D sparse convolution, making it easier to converge. The classification probabilities from both branches are multiplied finally. The utilization of such shared class-agnostic classification across domains also facilitates the model in learning general category knowledge in the 3D domain.

**Open-vocabulary extension.** Introducing language embeddings into classification enables our OneDet3D to be easily extended to the open-vocabulary setting, benefiting from the generalization ability of text to unseen categories. To further ensure category scalability, we follow [23] by first performing large-scale vocabulary inference on 2D images using a pre-trained 2D open-vocabulary detector [54], then projecting the obtained 2D boxes into 3D space to obtain 3D pseudo labels with an expanded vocabulary. With such large-vocabulary pseudo labels for multi-domain joint training, the

Table 2: **The performance of OneDet3D for closed-vocabulary 3D object detection.** OneDet3D is joint training on these four datasets, and can conduct inference on them with the same model architecture and only one set of parameters. $AP_e$, $AP_m$, $AP_h$ denote the AP metric on easy, moderate, hard subsets separately, and $AP^{KIT}$ denotes the AP metric computed the same as KITTI. Gray cells indicate that the method original papers report results on this dataset. We re-implement previous methods on other datasets and "-" indicates that the method fails to converge.

| Method | SUN RGB-D | | ScanNet | | KITTI | | | nuScenes | |
|---|---|---|---|---|---|---|---|---|---|
| | $AP_{25}$ | $AP_{50}$ | $AP_{25}$ | $AP_{50}$ | $AP_e$ | $AP_m$ | $AP_h$ | AP | $AP^{KIT}$ |
| *single-dataset training* | | | | | | | | | |
| VoteNet [25] | 57.7 | 35.7 | 58.6 | 33.5 | 60.1 | 46.2 | 31.2 | - | - |
| H3DNet [51] | 60.1 | 39.0 | 67.2 | 48.1 | 37.6 | 29.8 | 26.9 | - | - |
| GroupFree [21] | 63.0 | 45.2 | 69.1 | 52.8 | 72.0 | 58.4 | 55.5 | 10.2 | 8.6 |
| FCAF3D [30] | 63.8 | 48.2 | 70.7 | 56.0 | 82.3 | 70.5 | 68.2 | 46.5 | 45.5 |
| SECOND [45] | - | - | - | - | 90.7 | 79.8 | 75.7 | 58.4 | 59.2 |
| PointPillar [16] | - | - | - | - | 88.5 | 79.3 | 76.3 | 73.1 | 74.3 |
| PointRCNN [34] | 46.5 | 34.7 | 45.7 | 37.1 | 91.7 | 80.4 | 79.8 | 25.5 | 26.7 |
| Part-$A^2$ [35] | - | - | - | - | 91.7 | 82.4 | 80.2 | - | - |
| PV-RCNN [32] | - | - | - | - | 92.6 | 84.8 | 82.7 | 76.0 | 76.9 |
| CenterPoint [47] | 18.9 | 4.2 | 15.6 | 3.3 | 86.9 | 75.5 | 71.7 | 80.0 | 80.5 |
| VoxelNeXt [6] | 18.1 | 4.8 | 15.4 | 3.4 | 87.5 | 77.4 | 75.1 | 80.0 | 80.6 |
| UVTR [17] | 55.0 | 33.2 | 56.0 | 31.5 | 84.2 | 72.3 | 69.8 | 80.6 | 81.2 |
| **OneDet3D (ours)** | 63.2 | 48.7 | 69.9 | 55.1 | 91.8 | 82.4 | 80.0 | 80.2 | 80.9 |
| *multi-dataset training* | | | | | | | | | |
| **OneDet3D (ours)** | **65.0** | **51.3** | **70.9** | **56.2** | **92.8** | 84.2 | 82.3 | **81.0** | **81.8** |

generalization ability to novel categories can be boosted. The multi-dataset training manner makes it possible to comprehensively utilize different types of data from various domains, thus is quite suitable for the open-vocabulary setting. Through such open-vocabulary extension, OneDet3D can generalize to unseen categories. As a result, OneDet3D can generalize across various domains, categories, and scenes, thus can be considered to possess the capability of universal 3D object detection.

## 5  Experiments

In this section, we demonstrate the one-for-all ability of our OneDet3D through extensive experiments. Close-vocabulary and open-vocabulary 3D object detection experiments are both conducted. We mainly conduct multi-dataset joint training on SUN RGB-D [36], ScanNet [8], KITTI [11], and nuScenes [2] datasets, and utilize S3DIS [1] and Waymo [37] for unseen domains in cross-dataset experiments. We implement OneDet3D with mmdetection3D [7], and train it with the AdamW [22] optimizer. We use the 0.01m voxel size for indoor datasets and the 0.05m voxel size for outdoor ones. Besides this, other architecture-related hyper-parameters are all the same for different datasets. During multi-dataset training, the attribute channel size is set to the least common multiple of the attribute dimensions from the different datasets, which is 6-dim. The attributes of the point clouds from different datasets are repeated accordingly to match this unified channel size.

### 5.1  Closed-Vocabulary 3D Object Detection

We first conduct multi-dataset joint training on all the above mentioned four datasets, and perform closed-vocabulary inference. Specifically, for the SUN RGB-D dataset, we perform 3D detection on 10 classes, for ScanNet it's 18 classes, while for the KITTI and nuScenes datasets, we focus on the performance of the "car" category. The results are listed in Tab. 2. It can be seen that even in the traditional single-dataset training-and-testing paradigm, existing 3D detectors can only conduct detection in a specific domain. Indoor 3D detectors can only operate on indoor point clouds, while the majority of existing outdoor 3D detectors can only work on one of the KITTI and nuScenes datasets because of their differences in sparsity and scenes. In comparison, our model can directly perform training and inference on these different domain point clouds. After multi-dataset joint training, OneDet3D can perform 3D detection on all domain point clouds with only *one set of parameters*. The performance surpasses that of most existing methods trained and tested using single-dataset training and inference. For instance, on the SUN RGB-D dataset, OneDet3D achieves the 65.0%

$AP_{25}$, surpassing FCAF3D by 1.2%. On the outdoor KITTI dataset, OneDet3D performs comparably to PV-RCNN, and on nuScenes, its AP surpasses existing methods such as VoxelNeXt and UVTR. Moreover, after multi-domain joint training, the performance of OneDet3D exceeds its own from single-dataset training. On the SUN RGB-D and KITTI datasets, multi-dataset joint training brings a 1.8% improvement for both. This demonstrates that even with significant differences, OneDet3D can learn universal 3D detection knowledge from these diverse point clouds. The necessity of multi-domain joint training and the effectiveness of our OneDet3D can thus be demonstrated.

**Comparison with more recent methods.** We further compare with some more recent 3D detectors and list the comparison in Tab. 3. As can be seen, these recent methods target at specific 3D scenes. They may outperform OneDet3D in those particular datasets, but AP tends to drop when the scene changes, especially when switching from outdoor to indoor. After multi-dataset training, due to the dataset-aware interference, AP on all datasets degrade severely. In such multi-dataset scenarios, OneDet3D still achieves the best. Besides, compared with Uni3D [48], which has provided a unified model for outdoor point clouds, OneDet3D provides a universal solution for all point clouds. As can be seen, even compared with these recent methods, OneDet3D is still the first universal 3D detec-

Table 3: **Comparison with more recent 3D object detectors for closed-vocabulary 3D object detection.** We re-implement these methods on the SUN RGB-D (SUN), ScanNet (Scan), KITTI (KIT) and nuScenes (nuS) datasets, and compare the $AP_{25}$, $AP_m$, AP metrics.

| Method | SUN | Scan | KIT | nuS |
|---|---|---|---|---|
| *single-dataset training* | | | | |
| VoxelNeXt [6] | 18.1 | 15.4 | 77.4 | 80.0 |
| FSD v2 [10] | 25.3 | 29.1 | 75.6 | 82.1 |
| SAFDNet [49] | 12.9 | 11.6 | 80.3 | 84.7 |
| *multi-dataset training* | | | | |
| VoxelNeXt [6] | 8.3 | 9.9 | 68.4 | 71.0 |
| FSD v2 [10] | 13.6 | 12.9 | 60.1 | 72.4 |
| SAFDNet [49] | 3.2 | 1.9 | 38.7 | 70.4 |
| Uni3D [48] | 9.7 | 5.6 | 75.2 | 76.7 |
| **OneDet3D (ours)** | **65.0** | **70.9** | **84.2** | **80.9** |

tor that can generalize across various point clouds. The main reason is that the specific designs of these detection heads, such as vote-based methods or BEV detection, are influenced by the structure and content of point clouds and thus are only applicable to outdoor scenes. Additionally, these methods lack designs to address multi-dataset interference, resulting in performance degradation across all datasets during multi-dataset joint training. In contrast, the anchor-free detection head of our OneDet3D is more versatile for both indoor and outdoor scenes. Furthermore, domain-aware partitioning and language-guided classification can alleviate multi-dataset interference. Therefore, our approach provides a more universal solution for 3D detection.

## 5.2 Open-Vocabulary 3D Object Detection

Table 4: **The performance of OneDet3D on the SUN RGB-D and ScanNet dataset for open-vocabulary 3D object detection.** OneDet3D is jointly training on these two indoor datasets. The utilized data are totally the same as CoDA, downloaded from CoDA officially released code.

| | Method | SUN RGB-D | | | ScanNet | | |
|---|---|---|---|---|---|---|---|
| | | $AP_{novel}$ | $AP_{base}$ | $AP_{all}$ | $AP_{novel}$ | $AP_{base}$ | $AP_{all}$ |
| *single-dataset training* | Det-PointCLIP [50] | 0.08 | 5.07 | 1.27 | 0.12 | 2.46 | 0.63 |
| | Det-PointCLIPv2 [56] | 0.12 | 4.83 | 1.24 | 0.11 | 1.79 | 0.47 |
| | 3D-CLIP [28] | 3.02 | 31.13 | 9.74 | 3.59 | 14.56 | 5.97 |
| | CoDA [3] | 6.65 | 39.70 | 14.55 | 6.02 | 22.01 | 9.48 |
| | **OneDet3D (ours)** | **11.01** | **42.14** | **18.45** | **13.78** | **34.59** | **18.29** |
| *multi-dataset training* | **OneDet3D (ours)** | **12.59** | **44.49** | **20.22** | **15.52** | **35.11** | **19.77** |

We then conduct open-vocabulary 3D object detection experiments with our OneDet3D. In our experiments, we refer to the setting in CoDA, where SUN RGB-D involves 46 classes and ScanNet involves 60 classes in total. Their top 10 classes are used as base categories. For multi-dataset joint training, we combine the base categories from both datasets to form a union, resulting in a total of 16 base categories, with the rest as novel categories. We reproduce the results of existing methods under this new category division and list the comparison in Tab. 4. It's worth noting that for a fair comparison, we used exactly the same setting as CoDA, which utilizes a single-view image setting in ScanNet, slightly different from the above closed-vocabulary setting. It can be seen that the superiority of our method is more obvious here. On the SUN RGB-D dataset, we achieve the $AP_{novel}$ improvement of over 5.94% compared to CoDA. On the ScanNet dataset, we achieve the

Table 5: **The cross-domain performance of OneDet3D** on the indoor S3DIS dataset.

| method | Trained on | $AP_{25}$ | $AP_{50}$ |
|---|---|---|---|
| VoteNet | SUN RGB-D | 29.7 | 13.5 |
| | ScanNet | 34.9 | 21.7 |
| FCAF3D | SUN RGB-D | 41.8 | 21.7 |
| | ScanNet | 44.8 | 36.7 |
| **OneDet3D** | SUN RGB-D | 43.5 | 23.8 |
| | ScanNet | 48.5 | 38.8 |
| | SUN, ScanN | 52.6 | **42.6** |
| | SUN, ScanN, KIT, nuS | **53.5** | 42.4 |

Table 6: **The cross-domain performance of OneDet3D** on the outdoor Waymo dataset.

| method | Trained on | $AP^{3D}$ | $AP^{BEV}$ |
|---|---|---|---|
| PointPillar | KITTI | 3.1 | 7.3 |
| | nuScenes | 3.5 | 5.5 |
| PV-RCNN UVTR | KITTI | 4.1 | 9.2 |
| | nuScenes | 9.6 | 23.0 |
| **OneDet3D** | KITTI | 7.1 | 18.2 |
| | nuScenes | 17.1 | 40.6 |
| | KIT, nuS | 40.3 | 61.3 |
| | SUN, ScanN, KIT, nuS | **41.1** | **61.7** |

Table 7: **Ablation study of OneDet3D.** OneDet3D is joint training on the SUN RGB-D and KITTI datasets, and S3DIS is utilized for cross-domain evaluation. SP, CP, LGC is short for scatter partitioning, context partitioning and language-guided classification separately.

| | SP | CP | LGC | SUN RGB-D | | KITTI | | | S3DIS (unseen) | |
|---|---|---|---|---|---|---|---|---|---|---|
| | | | | $AP_{25}$ | $AP_{50}$ | $AP_e$ | $AP_m$ | $AP_h$ | $AP_{25}$ | $AP_{50}$ |
| *single-dataset training* | | | | 63.2 | 48.7 | 91.8 | 82.4 | 80.0 | 43.5 | 23.8 |
| *multi-dataset training* | | | | 61.9 | 45.9 | 91.1 | 81.4 | 79.8 | 41.4 | 22.6 |
| | ✓ | | | 63.2 | 48.9 | 91.8 | 82.9 | 80.5 | 44.2 | 25.9 |
| | ✓ | ✓ | | 63.9 | 49.5 | 91.9 | 83.1 | 80.6 | 45.8 | 27.6 |
| | ✓ | ✓ | ✓ | **64.4** | **49.9** | **92.2** | **83.5** | **80.8** | **47.8** | **29.0** |

15.52% $AP_{novel}$, surpassing CoDA by even more than 9%. This strongly demonstrates that our OneDet3D not only generalizes well across domains but also exhibits strong generalization ability at the category level. Compared to single-dataset training, multi-dataset training brings the over 1% improvement. This is because the utilization of multiple datasets enables to learn richer knowledge about categories. This experiment validates the universal ability of OneDet3D in terms of categories, which demonstrates its basic capability for universal 3D object detection.

## 5.3 Cross-Domain 3D Object Detection

We further conduct cross-domain 3D detection experiments, with S3DIS and Waymo as new domains for inference. The comparison results are listed in Tab. 5 and Tab. 6. It can be observed that on S3DIS, in single-dataset training, our method already outperforms existing methods. This is because our language-guided classification better alleviates category conflicts. The performance is slightly better when trained on ScanNet because the ScanNet domain is more similar to S3DIS. After training on both datasets, the cross-domain AP on S3DIS improves by more than 4%, indicating the model ability to integrate information from both domain datasets. Furthermore, with the introduction of two outdoor datasets, $AP_{25}$ improves by 0.9%, with $AP_{50}$ remaining stable. This demonstrates that our OneDet3D can learn from such highly different domain point clouds for enhancement in cross-domain 3D detection. In outdoor point clouds, this is even more pronounced. KITTI is relatively similar to Waymo but only contains small-scale point clouds, while nuScenes is larger-scale but exhibits a larger domain gap. Training separately on these two datasets thus only yields limited cross-dataset $AP^{3D}$ on Waymo. In comparison, through multi-dataset training, the model can utilize the characteristics of both, resulting in a substantial 23.1% improvement. This demonstrates the generalization capability of our method to unseen domains and further validates the necessity of multi-dataset training.

## 5.4 Ablation Study

We finally conduct ablation study in this subsection and list the results in Tab. 7 to evaluate our designs. Here we conduct multi-dataset training on the SUN RGB-D and KITTI datasets, and utilize S3DIS here for cross-domain evaluation. We also list the single-dataset training results as a reference. As can be seen, although our model architecture allows for multi-dataset joint training, directly multi-dataset training results in decreased AP on both seen and unseen domains, because of the interference problem. After introducing scatter partitioning, which alleviates interference among multiple domains during regularization, the model performance essentially matches and slightly exceeds that of single-dataset training. Then, through context partitioning, the detector can

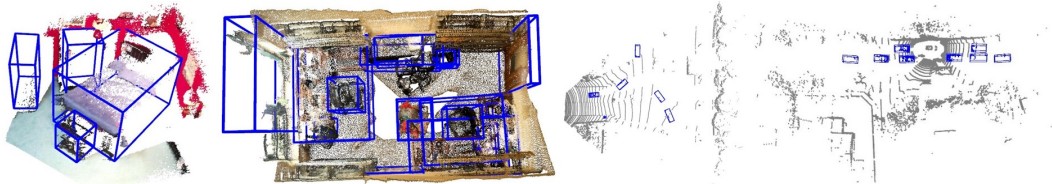

Figure 4: **The visualized results of OneDet3D** on the indoor SUN RGB-D, ScanNet and outdoor KITTI, nuScenes datasets separately.

selectively learn the corresponding global context of different domain point clouds, leading to further AP improvement. Especially for the indoor domain, the partitioning of global context learning enables multi-domain joint training AP to surpass those of single-domain training. Finally, with language-guided classification, the effect of category conflict can be mitigated. Since the category conflict problem between SUN RGB-D and KITTI is not severe, the performance improvement on these two datasets is relatively moderate. The about 0.5% AP improvement here is primarily because of the common class-agnostic classification. In cross-dataset experiments on S3DIS, language embeddings contribute to a more AP increase, more than 2% improvement. This is mainly because of the more substantial overlap in categories between SUN RGB-D and S3DIS, making language embeddings more effective for these two domains. Such the ablation study thus demonstrates the necessity of these designs to address the interference issues for multi-dataset joint training.

We provide visualized results from our OneDet3D in Fig. 4. It can be seen that no matter for indoor or outdoor point clouds from different domains, OneDet3D can perform 3D detection effectively using only one set of parameters. This further demonstrates its effectiveness and universal ability.

## 6 Conclusion

In this paper, we propose OneDet3D, a universal point cloud based 3D object detector that can generalize across various domains, categories, and scenes with only one set of parameters. The fully sparse structure and the anchor-free detection head serve as the basic model architecture. With partitioning in scatter and context, together with the language-guided classification, the interference caused by point clouds and categories can be alleviated. Extensive experiments demonstrate the strong one-for-all ability of OneDet3D. For the first time, we implement various scenarios and requirements of 3D object detection within a unified framework. This demonstrates that our OneDet3D has learned general 3D representations through multi-domain joint training, thus basically realizing the demands of universal 3D object detection and 3D foundation models. We believe that our research will stimulate following research along the universal computer vision direction in the future.

## Acknowledgement

This work is supported by Tsinghua University Toyota Joint Research Center for AI Technology of Automated Vehicle (TTAD 2024-07), National Natural Science Foundation of China (No. 62201484), HKU Startup Fund, and HKU Seed Fund for Basic Research.

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

# Appendix

In the appendix, we include more dataset introduction and implementation details. We also conduct more ablation studies to illustrate the reasonableness of our designs. More visualized examples are provided to further demonstrate the effectiveness of our OneDet3D.

## A Datasets and Implementation Details

**SUN RGB-D** [36]. SUN RGB-D is a single-view indoor dataset with 5,285 training and 5,050 validation scenes, annotated with 10 classes and oriented 3D bounding boxes. The point clouds within it are from converting RGB-D camera results. We utilize the 0.01m grid size for voxelization. During training, we randomly flip the input data along the $x$ axis, randomly sample 10,000 points, and apply global translation, rotation, scaling for data augmentation. We utilize the $AP_{25}$ and $AP_{50}$ metrics for evaluation.

**ScanNet** [8]. The ScanNet V2 dataset contains 1,201 reconstructed training scans and 312 validation scans, with 18 object categories for axis-aligned bounding boxes. The point clouds within it are from reconstructing from a series of multi-view images. We also utilize the 0.01m grid size for voxelization. During training, we randomly flip the input data along both the $x$ and $y$ axis, randomly sample 10,000 points, and apply global translation, rotation, scaling for data augmentation. We utilize the $AP_{25}$ and $AP_{50}$ metrics for evaluation. For the open-vocabulary setting, we adopt the same setting as CoDA [3], dividing the large panoramic scene from the original ScanNet V2 into several smaller point cloud scenes, each corresponding to a single-view image. There are 47,841 training samples and 4,886 validation samples ultimately for the open-vocabulary setting.

**KITTI** [11]. The KITTI dataset consists of 7,481 LiDAR samples for its official training set, and we split it into 3,712 training samples and 3,769 validation samples for training and evaluation. We only utilize the "car" category for training and evaluation. During training, we also adopt van class objects as car objects. The data augmentation operations are basically the same as previous outdoor 3D detectors like [9]. For the ground-truth sampling augmentation, we sample at most 20 cars from the database. 18000 points are randomly sampled at the training time. The predicted car objects are filtered at the threshold of 0.6 after inference. We utilize the $AP_{70}$ metric under 40 recall positions on the "car" category for evaluation.

**nuScenes** [2]. Compared to the KITTI dataset, the nuScenes dataset covers a larger range, with 360 degrees around the LiDAR instead of only the front view. Its point clouds are also more sparse (with 32-beam LiDAR compared to the KITTI 64 beams). We train on the 28,130 frames of samples in the training set and evaluate on the 6,010 validation samples. We do not predict the velocities or attributes of objects, to keep consistent with other datasets. We only utilize the "car" category for training and evaluation. For the ground-truth sampling augmentation, we sample at most 5 cars from the database. We utilize its official AP metric, averaging over match thresholds of 0.5, 1, 2, 4 meters. We also utilize the KITTI AP metric for evaluation.

**S3DIS** [1]. S3DIS consists of 3D scans from 6 buildings, 5 object classes annotated with axis-aligned bounding boxes. We use the official split and perform cross-dataset evaluation of our method on 68 rooms from Area 5. Its point clouds are also from reconstructing multi-view images. We utilize the $AP_{25}$ and $AP_{50}$ metrics for evaluation.

**Waymo** [37]. We utilize the Waymo dataset for cross-dataset evaluation. We evaluate on its 39,987 samples of its validation set. We also only evaluate on the "car" category (*i.e.*, the "vehicle" category). We utilize the KITTI metric, $AP_{70}$ under 40 recall positions in both 3D and BEV, for evaluation.

**Implementation details.** We implement OneDet3D with mmdetection3D [7]. We utilize the 3D sparse convolution based ResNet50 [15] as backbone, together with FPN [18] for feature extraction. We train the model with the AdamW [22] optimizer for 20 epochs. The initial learning rate is set to 0.0001 and is updated in the cyclic manner. Point clouds from different datasets are sampled uniformly. We remove the ground-truth sampling augmentation at the last 2 epochs.

Table 8: **Ablation study about the design details of context partitioning on the SUN RGB-D, KITTI and S3DIS datasets.** OneDet3D is joint training on the SUN RGB-D and KITTI datasets, and S3DIS is utilized for cross-domain evaluation. CP is short for context partitioning.

| | method | SUN RGB-D | | KITTI | | | S3DIS (unseen) | |
| --- | --- | --- | --- | --- | --- | --- | --- | --- |
| | | $AP_{25}$ | $AP_{50}$ | $AP_e$ | $AP_m$ | $AP_h$ | $AP_{25}$ | $AP_{50}$ |
| *single-dataset training* | - | 63.2 | 48.7 | 91.8 | 82.4 | 80.0 | 43.5 | 23.8 |
| *multi-dataset training* | no CP | 63.7 | 48.9 | 91.8 | 83.2 | 80.6 | 45.3 | 26.9 |
| | CP for indoor and outdoor | 62.3 | 47.6 | 88.4 | 78.6 | 75.6 | 42.5 | 31.2 |
| | CP for indoor only | **64.4** | **49.9** | **92.2** | **83.5** | **80.8** | **47.8** | **29.0** |

Table 9: **Ablation study about the design details of language-guided classification on the SUN RGB-D, KITTI and S3DIS datasets.** OneDet3D is joint training on the SUN RGB-D and KITTI datasets, and S3DIS is utilized for cross-domain evaluation. The listed method is about the operation used for classification.

| | method | SUN RGB-D | | KITTI | | | S3DIS (unseen) | |
| --- | --- | --- | --- | --- | --- | --- | --- | --- |
| | | $AP_{25}$ | $AP_{50}$ | $AP_e$ | $AP_m$ | $AP_h$ | $AP_{25}$ | $AP_{50}$ |
| *single-dataset training* | 3D sparse conv | 63.2 | 48.7 | 91.8 | 82.4 | 80.0 | 43.5 | 23.8 |
| *multi-dataset training* | 3D sparse conv | 63.9 | 49.5 | 91.9 | 83.1 | 80.6 | 45.8 | 27.6 |
| | CLIP embeddings (frozen) | 60.2 | 44.8 | 90.7 | 81.4 | 79.5 | 25.8 | 19.2 |
| | CLIP embeddings (trainable) | 64.0 | 49.5 | 91.8 | 83.2 | 80.8 | 46.0 | 27.7 |
| | 3D sparse conv + CLIP embeddings (frozen) | **64.4** | **49.9** | **92.2** | **83.5** | **80.8** | **47.8** | **29.0** |

## B  More Ablation Study

**Context partitioning.** We discuss the design details about partitioning the learning of global context and list such ablation study in Tab. 8. As can be seen, without the partitioning, the model can already achieve the performance through multi-dataset training that is better than the single-dataset training baseline, partially due to our design of scatter partitioning. If we apply context partitioning for both indoor and outdoor point clouds, the 3D detection AP decreases on all datasets. Especially for the outdoor KITTI dataset, $AP_m$ decreases by 3.8%. This is because global information in outdoor scenes tends to be disruptive, given the excessive background points and the fact that foreground objects occupy only a small portion of the scene. In comparison, introducing context partitioning only for indoor point clouds can lead to a more than 1% AP improvement, and the cross-dataset AP improvement is 2.5%. This is because indoor scenes are smaller and the object distribution is more crowded, making global context relatively more important thus contributing to performance improvement. Therefore, the rationality of our context partitioning design is validated.

**Language-guided classification.** We then discuss the design details about our language-guided classification and list the related 3D detection results in Tab. 9. The anchor-free detection head typically employs 3D sparse convolution for final classification. However, this approach struggles to address the category conflict issue among different domains. Especially during cross-dataset evaluation, differences in category definitions across datasets and the potential new categories in new domains can all restrict the model performance. Utilizing language embeddings as the classification layer can help mitigate this issue, due to the generalization ability of the text modality. However, directly using CLIP embeddings for classification results in a decrease in detection AP across all datasets. This is because a frozen fully connected layer impedes gradient backpropagation in the fully sparse convolution structure. Instead, when CLIP embeddings are trainable instead of frozen, the achieved AP can be comparable to or slightly surpass that of 3D sparse convolution. However, in this way, CLIP embeddings are not available at the inference time, and the model thus cannot generalize to new domains, especially new categories during inference.

In comparison, what we utilize is the combination of 3D sparse convolution and frozen CLIP embeddings. 3D sparse convolution is utilized for class-agnostic classification, and is shared among all domains. The frozen CLIP embeddings are utilized for class-specific classification, and can benefit the model with the help of the text modality. As a result, both detection AP from the indoor and outdoor domains can be boosted. The cross-dataset AP on S3DIS increases the most, a 2% $AP_{25}$ improvement, because the category conflict problem can be alleviated. The effectiveness of our designs in language-guided classification is thus demonstrated.

Table 10: **Ablation study about the use of classification loss on the SUN RGB-D, KITTI and S3DIS datasets.** OneDet3D is joint training on the SUN RGB-D and KITTI datasets, and S3DIS is utilized for cross-domain evaluation.

| method | SUN RGB-D | | KITTI | | | S3DIS (unseen) | |
|---|---|---|---|---|---|---|---|
| | $AP_{25}$ | $AP_{50}$ | $AP_e$ | $AP_m$ | $AP_h$ | $AP_{25}$ | $AP_{50}$ |
| focal loss [19] | not converged | | | | | | |
| soft focal loss, $IoU_{3D}$ | 57.6 | 41.2 | 86.3 | 76.2 | 73.9 | 35.9 | 18.2 |
| soft focal loss, $IoU_{de}$ [41] | 62.1 | 48.4 | 88.5 | 78.9 | 76.3 | 43.2 | 23.8 |
| soft focal loss, $IoU_{BEV}$ | **64.4** | **49.9** | **92.2** | **83.5** | **80.8** | **47.8** | **29.0** |

**Classification loss.** We then analyze the choice of the loss function for classification, and list the comparative results in Tab. 10. It can be observed that if we utilize focal loss for classification, which is widely utilized in the anchor-free detection head, the model cannot converge. This problem appears after the introduction of the language-guided classification. Since the class-specific classification is implemented by CLIP embeddings, which are frozen during training, the model requires a stronger optimization way for learning. The usual used focal loss thus becomes insufficient in this task. Utilizing soft focal loss, with IoU as the soft target can introduce positional information in classification, thus alleviating this problem. When using $IoU_{3D}$ as the soft target, as $IoU_{3D}$ is relatively hard to optimize, especially coupled with the classification task, the performance is still limited. The use of decoupled IoU proposed in [41] can alleviate this problem, by decoupling IoU calculation in the $xy$ plane and the $z$ axis. However, this decoupling is still not enough in our task, because the frozen CLIP embeddings make optimization more challenging. Therefore, we directly disregard the $z$ direction here, and utilize $IoU_{BEV}$ here. The positional information in the $xy$ plane can already provide sufficient supervision signals for classification, and disregarding the $z$ direction also makes optimization easier. Ultimately, the model can conduct supervised learning with the language-guided classification, and achieve satisfying results on both indoor and outdoor point clouds.

## C    Visualized Results

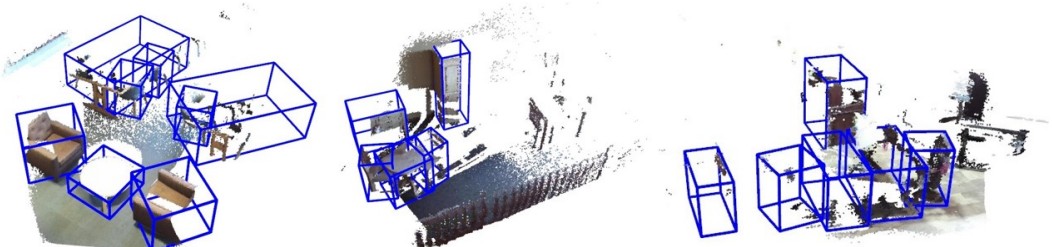

Figure 5: **The visualized results of OneDet3D** on the indoor SUN RGB-D dataset.

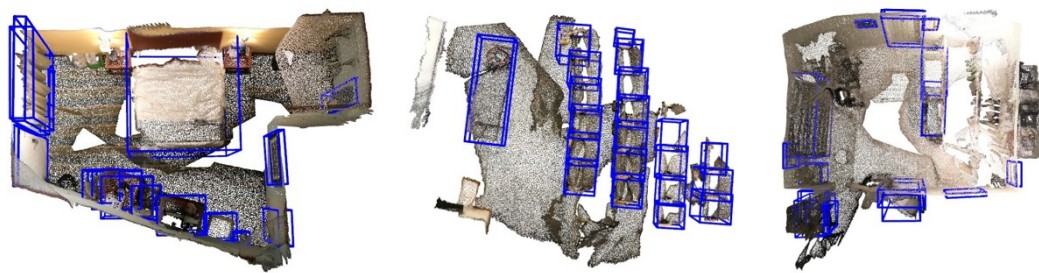

Figure 6: **The visualized results of OneDet3D** on the indoor ScanNet dataset.

We further provide more visualized results on the indoor SUN RGB-D dataset (Fig. 5), indoor ScanNet dataset (Fig. 6), outdoor KITTI dataset (Fig. 7), and outdoor nuScenes dataset (Fig. 8). OneDet3D obtains satisfying detection results on all these four datasets, with only one set of parameters. Our method can accurately detect objects in various domain point clouds, ranging from indoor crowded

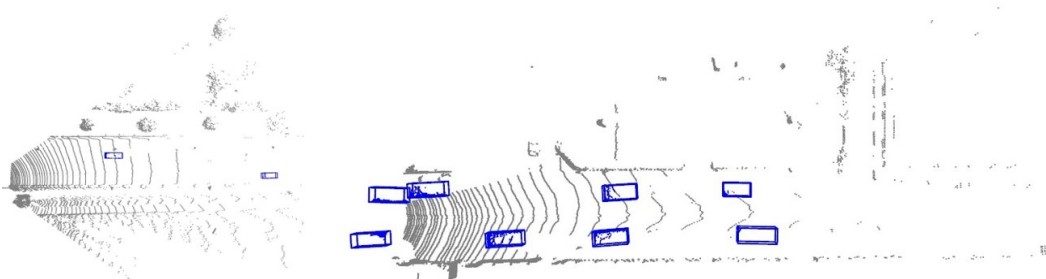

Figure 7: **The visualized results of OneDet3D** on the outdoor KITTI dataset.

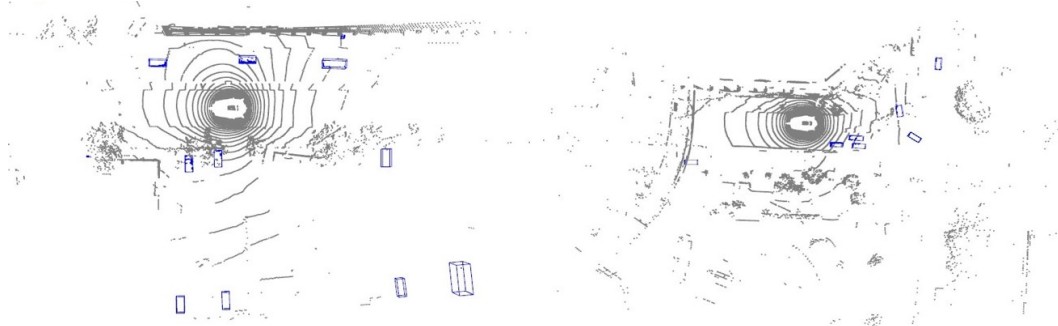

Figure 8: **The visualized results of OneDet3D** on the outdoor nuScenes dataset.

scenes with overlapping objects to outdoor scenes with small-sized objects. This further demonstrates its effectiveness and universality.

**Limitation and Potential Negative Social Impacts.** Our approach addresses the multi-domain joint training issue in point clouds, aiming to achieve a universal 3D detector. Currently, it focuses on supervised learning. Its utilized point clouds should be fully-annotated, with both categories and 3D boxes. In the future, we plan to incorporate weakly-labeled or unlabeled data to enable the model to conduct larger-scale learning. Additionally, since our model can utilize various sources of point clouds for training, the misuse of collected data could potentially lead to negative social impacts.

