# OpenReview forum: "One for All: Multi-Domain Joint Training for Point Cloud Based 3D Object Detection"
_NeurIPS.cc/2024/Conference — NeurIPS 2024 poster_

### Official Review · Reviewer_HkhJ · 2024-07-01

**Soundness:** 4
**Presentation:** 3
**Contribution:** 4
**Rating:** 6
**Confidence:** 5

**Summary:**

This paper proposes OneDet3D, a universal one-for-all model that addresses 3D detection across various domains, including both indoor and outdoor scenes. It tackles two primary issues: data-level interference caused by differences in the point clouds themselves and category-level interference caused by label conflicts among categories across different domains.

**Strengths:**

1.	The paper introduces a universal one-for-all 3D detection model applicable across different domains, using only one set of parameters.
2.	The proposed method demonstrates superior performance and remarkable generalization ability compared to existing state-of-the-art approaches.

**Weaknesses:**

1.	In lines 60-62, the authors claim that 3D sparse convolution is better than point-based feature extractors due to its robustness to domain gaps. Could the authors elaborate on this in detail?
2.	Regarding the domain router, how is the domain label n obtained? Does data from the same dataset share the same domain label?
3.	In Table 1, different datasets have different views. I wonder if any design could be implemented to tackle this difference for better generalization ability.
4.	In Equation 1, experiments regarding the hyperparameters α and c are missing. The values of these hyperparameters are not mentioned.
5.	In Table 2, the best performance achieved by existing methods is not highlighted in bold.
6.	Table 4 underlines the second-best performance, while Table 5 does not. Table 4 underlines the second-best performance, while Table 5 does not. It would be beneficial to maintain consistency.

**Questions:**

Please refer to the weakness section.

---

> ### Author Rebuttal · Authors · 2024-08-06
>
> **Q1: In lines 60-62, the authors claim that 3D sparse convolution is better than point-based feature extractors due to its robustness to domain gaps. Could the authors elaborate on this in detail?**
>
> A1:
>
> Table 3-1: Comparison with point-based feature extractor
> |      |  SUN RGB-D  | ScanNet | KITTI | nuScenes |
> | :--: | :--: | :--: | :--: | :--: |
> |point-based feature extractor|28.3|27.2|15.5|12.1|
> |3D sparse convolution based feature extractor|65.0|70.9|84.2|80.9|
>
> Point-wise feature extractors typically exploit metric space distances to learn local features. They extract features from point clouds by leveraging geometric characteristics through operations such as sampling and grouping. These operations heavily rely on the geometric information and metric space distances inherent in point clouds. Consequently, when dealing with point clouds from different domains, especially between indoor and outdoor scenes, the vast differences in scene size, object dimensions, and other aspects can severely disrupt the ability of point-wise feature extractors to learn generalizable features. In contrast, 3D sparse convolution operates directly on the voxelized feature space, making it robust to the difference in point clouds. This makes it better suited for the requirements of multi-domain joint training.
>
> We also conduct the comparative experiments and list the results in the above Tab. 3-1. As can be seen, 3D sparse convolution performs significantly better than the point-wise feature extractor. This further demonstrates the effectiveness of 3D sparse convolution when it comes to the large domain gap in point clouds, compared to point-wise feature extractors.
>
> **Q2: Regarding the domain router, how is the domain label n obtained? Does data from the same dataset share the same domain label?**
>
> A2: We will number the n datasets from 0 to n−1, assigning these numbers as the dataset classification labels. Data from the same dataset will share the same domain label, which will then be used for classification within the domain router.
>
> **Q3: In Table 1, different datasets have different views. I wonder if any design could be implemented to tackle this difference for better generalization ability.**
>
> A3: Both the model architecture and method designs support the better generalization ability for view difference. On the one hand, our OneDet3D adopts a fully sparse architecture. This architecture operates directly on points without requiring dense features with the fixed sizes, making it more flexible and robust in feature extraction when dealing with datasets from different views. On the other hand, in domain-aware partitioning, scatter partitioning is used to partition the normalization layers of point clouds from different domains. This approach prevents data-level interference between point clouds from different domains, thereby effectively addressing the issue of different views.
>
> **Q4: In Equation 1, experiments regarding the hyperparameters $\alpha$ and $c$ are missing. The values of these hyperparameters are not mentioned.**
>
> A4:
>
> $\alpha$ is set to 0.25, and $\xi$ is set to 2.0. $c$ is not a hyperparameter and is just the binary target class label.
>
> To illustrate its reasonability, we also conduct the ablation study about these two hyperparameters on the SUN RGB-D dataset, as in the below Tab. 3-2 and Tab. 3-3. As can be seen, our model is relatively robust to the choice of hyperparameters and we utilize the optimal ones.
>
> Table 3-2: Hyperparameter analysis on $\alpha$
> |   $\alpha$   |  AP25 | AP50 |
> | :--: | :--: | :--: |
> |0.1|59.7|40.2|
> |0.2|62.8|47.1|
> |0.25|65.0|51.3|
> |0.3|63.6|48.9|
>
> Table 3-3: Hyperparameter analysis on $\xi$
> |   $\xi$   |  AP25 | AP50 |
> | :--: | :--: | :--: |
> |1.0|55.0|35.7|
> |1.5|63.2|48.3|
> |2.0|65.0|51.3|
> |2.5|64.3|29.4|
>
> **Q5: In Table 2, the best performance achieved by existing methods is not highlighted in bold.**
>
> A5: Thank you for your suggestion. We will highlight the best performance achieved by existing methods in Table 2 in bold.
>
> **Q6: Table 4 underlines the second-best performance, while Table 5 does not. It would be beneficial to maintain consistency.**
>
> A6: Thanks for your suggestion. We will underline the second-best performance for both in the final version to maintain consistency.

---

> > ### Comment · Reviewer_HkhJ · 2024-08-12
> >
> > Thank you for providing the additional experiments and detailed explanations in your rebuttal. It addressed my concerns effectively.

---

### Official Review · Reviewer_q3RX · 2024-07-11

**Soundness:** 2
**Presentation:** 2
**Contribution:** 3
**Rating:** 5
**Confidence:** 4

**Summary:**

This paper proposes OneDet3D, which is a multi-domain jointly trained point cloud object detector for universal 3D object detection. It is the first 3D detector that supports point clouds from both indoor and outdoor scenes simultaneously with only one set of parameters. The experiments are conducted on multiple indoor and outdoor benchmarks.

**Strengths:**

- The proposed techniques are simple yet effective with intuitive explanations.
- This work showcases that multi-domain training can achieve better performance than single-domain training, which is a valuable observation.

**Weaknesses:**

- The performance in Table 2 for the nuScenes dataset is strange, where AP is not aligned with the numbers reported by other methods.

- The sizes of the adopted datasets are quite different. Are any resampling strategies adopted? How to mix them up? The training scheme and implementation details should be added, such as training schedule, data augmentation, and so on.

- The attributes of point clouds are different. For example, indoor datasets may contain RGB information and outdoor data such as Waymo may contain elongation and timestamp information. How to deal with the different channel sizes? A lot of details are ignored by the authors.

- There are many basic grammar issues. For example, "is joint training on" should be "is jointly trained on".

- I notice the authors claim and adopt fully sparse architecture. However, there is no comparison or discussion with previous work with fully sparse structures such as [1 - 4]. What are the differences between the proposed architecture and theirs? Do the issues they addressed still exist?

- Sec. 3.3 is hard to follow and should be rewritten with more details and sufficient explanation. (1) The description of "we utilize language vocabulary embeddings from CLIP" should be more detailed. Does the network predict CLIP embedding? How to supervise?  (2) Why is it necessary to convert sparse point features to dense features in Sec. 3.3?  (3) What is the specific definition of the dense features?  (4) Why is the back propagation unfeasible? (5) How many classification branches does it have? What does the "both branches" in L215 mean?

- Lack of comparison with other similar methods such as [5,6] in Table 4 & 5. \

[1] Fully Sparse 3d object detection \
[2] Voxelnext: Fully sparse voxelnet for 3d object detection and tracking \
[3] FSD V2: Improving Fully Sparse 3D Object Detection with Virtual Voxels \
[4] SAFDNet: A Simple and Effective Network for Fully Sparse 3D Object Detection \
[5] ST3D: Self-training for Unsupervised Domain Adaptation on 3D Object Detection \
[6] Uni3D: A Unified Baseline for Multi-dataset 3D Object Detection

**Questions:**

See the weakness part. I would like to increase the score if my concerns are well addressed.

**Limitations:**

The authors do not discuss the limitations and social impacts, which should be added.

---

> ### Author Rebuttal · Authors · 2024-08-06
>
> **Q1: The performance in Table 2 for the nuScenes dataset is strange.**
>
> A1: The reason is that we train and evaluate only on the car category of the nuScenes dataset. Since only the car class is involved in training, we do not need to use the CBGS sampler for class balance optimization during training, which significantly increases the number of training iterations. These factors cause our results to not be aligned with the numbers reported by other methods. For example, for the UVTR method, the original paper reports the AP of 60.9% for all 10 classes, while its AP for the car category is 84.8%. Without using the CBGS sampler, the car AP becomes 80.6%, which is the result we report in our paper. We will illustrate this in the final version.
>
> **Q2: Are any resampling strategies adopted? How to mix them up? The training scheme and implementation details should be added, such as training schedule, data augmentation, and so on.**
>
> A2: When mixing different datasets, we first perform dataset-wise uniform sampling across different datasets. Then, we sample individual point clouds from each dataset for training. Detailed explanations of the training scheme and implementation details are provided in the appendix (L460). The augmentations mainly include global translation, rotation, scaling, and ground-truth sampling augmentation. We train the model with the AdamW optimizer for 20 epochs. The initial learning rate is set to 0.0001 and is updated in a cyclic manner.
>
> **Q3: How to deal with the different channel sizes?**
>
> A3: During multi-dataset training, the attribute channel size is set to the least common multiple of the attribute dimensions from the different datasets, which is 6-dim. The attributes of the point clouds from different datasets are repeated accordingly to match this unified channel size.
>
> **Q4: There are many basic grammar issues.**
>
> A4: Thanks for your suggestions. We will correct them in the final version.
>
> **Q5: There is no comparison or discussion with previous work with fully sparse structures such as [1 - 4]. What are the differences between the proposed architecture and theirs? Do the issues they addressed still exist?**
>
> A5:
>
> Table 2-1: Comparison with previous work with fully sparse structures
> | |SUN RGB-D|ScanNet|KITTI|nuScenes|
> | :--: | :--: | :--: | :--: | :--: |
> |VoxelNeXt|8.3|9.9|68.4|71.0|
> |FSD v2|13.6|12.9|60.1|72.4|
> |SAFDNet |3.2|1.9|38.7|70.4|
> |OneDet3D |65.0|70.9|84.2|80.9|
>
> We compare our method with some existing fully sparse architectures, as shown above. These methods typically aim to achieve a more elegant and efficient network design. However, these methods are generally designed for outdoor scenes. Consequently, the specific designs of their detection heads, such as vote-based methods or BEV detection, are influenced by the structure and content of point clouds and thus are only applicable to outdoor scenes. Additionally, these methods lack designs to address multi-dataset interference, resulting in performance degradation across all datasets during multi-dataset joint training.
>
> In contrast, the anchor-free detection head of our OneDet3D is more versatile for both indoor and outdoor scenes. Furthermore, domain-aware partitioning and language-guided classification can alleviate multi-dataset interference. Therefore, our approach provides a more universal solution for 3D detection.
>
> **Q6: Sec. 3.3 is hard to follow and should be rewritten with more details and sufficient explanation.**
>
> A6:
> * We use the prompt "a photo of {name}" to extract language embeddings of the category names from different datasets using CLIP. These language embeddings are then used as parameters of the fully connected layer to perform the final classification, and are kept frozen during training. Our network does not need to predict such embeddings.
> * Since the language embeddings from CLIP are dense features, to utilize the fully connected layer with such embeddings for classification, it is necessary to convert the sparse point features to dense features.
> * Dense features refer to commonly used non-sparse features. They are expressed in the form of multi-dimensional matrix and stored as tensors within the network for computation.
> * Since the language embeddings from CLIP are kept frozen during training, the parameters in the fully connected layer for classification are frozen. Backpropagation is thus unfeasible in the fully connected layer. This makes the network training relatively difficult.
> * We have two classification branches. One branch is the frozen fully connected layer, utilizing CLIP embeddings. The other is a sparse convolution layer. This is trainable and is utilized for class-agnostic classification. “Both branches” in L215 thus denote these two branches.
>
> **Q7: Lack of comparison with other similar methods such as [5,6].**
>
> A7:
>
> Table 2-2: Comparison with Uni3D
> |      |  SUN RGB-D  | ScanNet | KITTI | nuScenes |
> | :--: | :--: | :--: | :--: | :--: |
> |Uni3D|9.7|5.6|75.2|76.7|
> |OneDet3D |65.0|70.9|84.2|80.9|
>
> * Uni3D: We compare with Uni3D in the above Tab. 2-2. It can be seen that Uni3D can only handle outdoor point clouds, while OneDet3D provides a universal solution for all point clouds.
> * ST3D: ST3D is designed for the unsupervised domain adaptation (UDA) task. Firstly, ST3D can only be applied to outdoor scenes and cannot be used for indoor scenes. Secondly, due to its focus on UDA, ST3D requires unlabeled target domain point clouds during training to perform cross-dataset experiments at test time. The trained model can only be used in the corresponding target domain, making this paradigm relatively inflexible. In contrast, our OneDet3D is more universal. It can be used in both indoor and outdoor scenes. Moreover, once trained, it can be directly applied to various scenes without requiring point clouds from the corresponding domain during training. Therefore, our model is more general and flexible.

---

> > ### Comment · Reviewer_q3RX · 2024-08-11
> >
> > Thanks so much for your response. It addressed most of my concerns. I appreciate the efforts to implement fully sparse methods such as voxelnext and fsdv2 in the indoor datasets, which are supposed to be added into the main paper. I would like to increase my score.

---

### Official Review · Reviewer_PVw6 · 2024-07-12

**Soundness:** 2
**Presentation:** 2
**Contribution:** 2
**Rating:** 4
**Confidence:** 4

**Summary:**

This manuscript introduces OneDet3D, a universal point cloud-based 3D object detector designed to address the challenges of multi-domain joint training. The primary motivation is to overcome the limitations of existing 3D detectors, which are typically trained and tested on single datasets, restricting their generalization across different domains. OneDet3D aims to provide a unified solution that can handle diverse indoor and outdoor scenes using a single set of parameters.

The authors claimed the following contributions of OneDet3D:

- Domain-Aware Partitioning: This technique aims to address data-level interference caused by differences in point cloud structures across domains. The parameters related to data scatter and context learning are partitioned and guided by a domain router, allowing the model to learn domain-specific features without increasing complexity.

- Language-Guided Classification: By incorporating text modality through CLIP embeddings, OneDet3D mitigates category-level interference among different datasets. This approach allows for better generalization to new domains and categories.

- Fully Sparse Architecture: The use of 3D sparse convolution and an anchor-free detection head makes the model robust to domain gaps and efficient in handling point clouds from various domains.

Experiments on datasets like SUN RGB-D, ScanNet, KITTI, and nuScenes demonstrate the effectiveness of OneDet3D. The model achieves promising performance in both closed-vocabulary and open-vocabulary settings, showing improvements over existing methods and strong generalization abilities.

**Strengths:**

(+) OneDet3D addresses the challenge in 3D object detection by introducing multi-domain joint training, which enhances the model's ability to generalize across various indoor and outdoor scenes. Such an endeavor is in line with the current research trend for point cloud 3D perception.

(+) The manuscript includes extensive experiments and evaluations on multiple datasets, showcasing the model's superior performance and generalization capabilities compared to some state-of-the-art methods.

(+) The overall OneDet3D framework seems standard and scalable; with more datasets and larger model sizes involved, the performance could be further improved.

**Weaknesses:**

(-) The overall OneDet3D framework is a combination of several previous approaches, which might not demonstrate a strong novelty in the related area:
- The domain router and context partitioning from Sec. 3.2 is similar to what [R1] and [R2] did for reducing domain differences.
- The scatter partitioning method in Sec. 3.2 is closely related to Uni3D [R3] (see their Sec. 3.3).
- The language-guided classification in Sec. 3.3 shares the same intuition with [R4] and [R2].

(-) The experimental comparisons could benefit from involving more recent 3D object detectors. For example, the latest model in Tab. 2 (UVTR) is from two years ago; while most of the other models are from 2020 (or even earlier).

(-) The overall elaboration of this manuscript deserves further improvements. Several claims are made without supporting evidence or references. Besides, most of the technical details are given in the text; having more graphical illustrations or algorithm flows could reduce the redundancy in the elaboration and help readers better understand the proposed method.

---
### References:
- [R1] Towards Universal Object Detection by Domain Attention. CVPR, 2019.
- [R2] Multi-Space Alignments Towards Universal LiDAR Segmentation. CVPR, 2024.
- [R3] Uni3D: A Unified Baseline for Multi-Dataset 3D Object Detection. CVPR, 2023.
- [R4] DaTaSeg: Taming A Universal Multi-Dataset Multi-Task Segmentation Model. NeurIPS, 2023.

**Questions:**

- **Q1:** As mentioned in Weakness 1, it is recommended to make a clearer comparison and have additional discussions with closely related works from existing literature. Putting the Related Work section behind the Introduction section and adding more detailed analyses and discussions with existing works can be beneficial. Including experimental comparisons and ablation studies with existing works, such as Uni3D, could further justify the effectiveness of the proposed OneDet3D.

- **Q2:** As mentioned in Weakness 2, supplementing the results with more recent single-dataset training approaches could further improve the comprehensiveness of the benchmark studies.

- **Q3:** As mentioned in Weakness 3, the manuscript could benefit from having more graphical illustrations or algorithm flows, instead of just plain text descriptions. This is also in line with the style of ML conferences.

- **Q4:** For the scatter partitioning method in Sec. 3.2: Since the dataset-specific statistics are used, how does OneDet3D handle a new point cloud with unknown statistics during inference?

- **Q5:** How does OneDet3D handle the class mapping differences using the language-guided classification in Sec. 3.3? For example, how to handle cases like the different definitions of `bicycle` and `bicyclist` (in KITTI and Waymo)?

**Limitations:**

The authors acknowledge several limitations in their approach: The current focus on supervised learning with fully annotated point clouds limits scalability. Future work should explore weakly labeled or unlabeled data to reduce reliance on extensive annotations.

Additionally, the manuscript lacks a detailed analysis of the computational cost associated with the proposed methods, which is crucial for assessing real-time application feasibility. What is more, there is a risk of overfitting to seen domains during multi-domain training. More experiments on unseen domains would strengthen the claims of generalization. The scalability and generalizability of OneDet3D to other types of sensors and different environmental settings have not been extensively discussed. Further research is needed to assess the model's robustness in diverse real-world scenarios.

---

> ### Author Rebuttal · Authors · 2024-08-06
>
> **Q1: It is recommended to make a clearer comparison and have additional discussions with closely related works**
>
> A1:
> * Our main contribution is that we propose **a universal 3D detector that can directly generalize across various indoor and outdoor point clouds**, once trained. Existing works only support either indoor or outdoor point clouds and cannot achieve cross-dataset, *especially cross-scene, generalization*. **We are the first** to implement a universal solution for all types of point clouds, which is our core novelty.
> * For this, we tackle multi-dataset joint training involving both indoor and outdoor point clouds. There exist significant differences in structure and content between indoor and outdoor point clouds, making this task highly challenging. In contrast, [R1] and [R4] deal with multi-dataset training with RGB images, [R2] and [R3] focus on outdoor-only multi-dataset training, where the discrepancies between different datasets are far less than those between indoor and outdoor point clouds. OneDet3D demonstrates that despite these substantial differences, 3D detection can still be addressed with a universal solution. This is a crucial advancement for generalization in the 3D domain.
> * Our designs also differ significantly from existing methods. [R1] and [R2] merely perform knowledge aggregation across different domains. In contrast, we design a domain router to direct information flow, better preventing data-level interference. [R3] uses dataset-aware channel-wise calculation for mean and variance in BN, while we decouple the scaling and shifting parameters. Compared with [R4], we also address structural optimization, using sparse convolution for class-agnostic and FC layers for class-specific classification. In a word, our methods are specifically designed to address the requirements of the challenging problem, making them fundamentally different from existing methods. These designs form a comprehensive system, spanning model architecture, method design, and training, making it *more than just a simple combination*.
> * We compare with Uni3D in the below Tab. 1-1. Uni3D can only handle outdoor point clouds, while OneDet3D provides a universal solution for all point clouds. We will include a discussion of these methods and reposition the related work section to make the discussions more comprehensive.
>
> Table 1-1: Comparison with Uni3D
> |      |  SUN RGB-D  | ScanNet | KITTI | nuScenes |
> | :--: | :--: | :--: | :--: | :--: |
> |Uni3D|9.7|5.6|75.2|76.7|
> |OneDet3D |65.0|70.9|84.2|80.9|
>
> **Q2: Supplementing the results with more recent single-dataset training approaches could further improve the comprehensiveness**
>
> A2:
>
> Table 1-2: Comparison with more recent methods
> | | | SUN RGB-D|ScanNet|KITTI|nuScenes|
> | :--: | :--: | :--: | :--: | :--: | :--: |
> |single-dataset training|VoxelNeXt [CVPR23]|18.1|15.4|77.4|80.0|
> | | FSD v2 [Arxiv 2308]|25.3|29.1|75.6|82.1|
> | | SAFDNet [CVPR24]|12.9|11.6|80.3|84.7|
> |multi-dataset training|VoxelNeXt [CVPR23]|8.3|9.9|68.4|71.0|
> | | FSD v2 [Arxiv 2308]|13.6|12.9|60.1|72.4|
> | | SAFDNet [CVPR24]|3.2|1.9|38.7|70.4|
> | |OneDet3D |65.0|70.9|84.2|80.9|
>
> These recent methods target at specific 3D scenes. They may outperform OneDet3D in those particular datasets, but AP tends to drop when the scene changes, especially when switching from outdoor to indoor. After multi-dataset training, due to the dataset-aware interference, AP on all datasets degrade severely. In such multi-dataset scenarios, OneDet3D still achieves the best. Even compared with these recent methods, OneDet3D is still the first universal 3D detector that can generalize across various point clouds.
>
> **Q3:  The manuscript could benefit from having more graphical illustrations or algorithm flows.**
>
> A3: We include a flowchart in Fig. 1 of the rebuttal document. We will incorporate this as an algorithm to provide a clearer explanation.
>
> **Q4: How does OneDet3D handle a new point cloud with unknown statistics during inference?**
>
> A4:
>
> Table 1-3: Cross-dataset performance on ScanNet
> | | trained on|AP25|
> | :--: | :--: | :--: |
> |VoteNet|SUN RGB-D|15.3|
> |FCAF3D|SUN RGB-D|26.1|
> |OneDet3D|SUN RGB-D|29.2|
> |OneDet3D|SUN RGB-D, KITTI|31.1|
>
> The most dataset-specific statistics, i.e., the mean and variance, are calculated based on the current batch of point clouds. For point clouds from the new domain, the mean and variance are computed according to the current data. We primarily partition the scaling and shifting parameters. For new domain point clouds, the domain router will calculate the domain probability to assess their similarity to existing domains. As is shown in Equ. 3 of our paper, the domain probability weights the outputs from different dataset-specific statistics, addressing the inference issue for new domains.
>
> We further train our model on SUN RGB-D and KITTI, and test it on ScanNet. The results show that our method remains effective for new point clouds with unknown statistics, because the weighted manner allows to select dataset-specific statistics most similar to the test data. This further validates the reasonability of our design.
>
> **Q5: How does OneDet3D handle the class mapping differences using the language-guided classification?**
>
> A5: Point clouds from different domains use their own CLIP embeddings for classification. They are independent in class-specific classification. As a result, point clouds from different domains can be classified independently, alleviating class mapping differences. Experiments show that despite the different definitions of "car" in KITTI and nuScenes, the performance improves after multi-domain joint training, demonstrating the effectiveness of language-guided classification.
>
> **Q6: The manuscript lacks a detailed analysis of the computational cost**
>
> A6:
>
> Table 1-4: Efficiency comparison
> | |FPS|
> | :--: | :--: |
> |CenterPoint|32.8|
> |UVTR|20.6|
> |OneDet3D|34.9|
>
> As can be seen, due to the fully sparse architecture, our network is highly efficient.

---

> ### Author Response · Authors · 2024-08-12
>
> Dear Reviewer PVw6:
>
> We thank you for the precious review time and valuable comments. We have provided corresponding responses and results, which we believe have covered your concerns. We hope to further discuss with you whether or not your concerns have been addressed. Please let us know if you still have any unclear parts of our work.

---

> ### Author Response · Authors · 2024-08-14
> **Looking forward to Feedback as Discussion Deadline Approaches**
>
> Thanks for your thorough reviews, which are very helpful to improving the quality of our paper. We apologize for any inconvenience caused, but as the deadline for discussion (Aug 13 11:59 pm AoE) draws near, we would like to provide an update on our progress.
>
> If you need further clarification or have additional questions, please don't hesitate to contact us. Again, we sincerely thank you for your time and effort in reviewing our paper.
>
> Thanks

---

### Author Rebuttal · Authors · 2024-08-06

We include the flowchart of our method in the PDF file here.

---

### Decision · Program_Chairs · 2024-09-25

**Decision:**

Accept (poster)

**Comment:**

The paper receives two positive and one negative ratings. Overall, the main concerns from the reviewers are some technical clarifications (e.g., how to handle unseen domains, dataset sampling/arrangement) and more comparisons with the prior work. After the rebuttal, two reviewers find that the major concerns are addressed well, where one reviewer upgraded the rating to borderline accept. For reviewer PVw6 who did not update the comment, the AC took a close look and find that all the major comments are well addressed in the rebuttal, including the technical comparisons with the prior work, more comparisons with 3D object detectors and cross-dataset setting with unseen domain. After reading the paper and reviews, the AC finds the merit of paper solving an important task for multi-domain 3D object detection, with well supported technical contributions and extensive experiments. Therefore, the AC recommends the acceptance decision and encourages the authors to incorporate the suggestion from the reviewers in the final version, as well as releasing the implementation to foster the development of this field.